# Poisson–Gamma Dynamical Systems

**Aaron Schein**
College of Information and Computer Sciences
University of Massachusetts Amherst
Amherst, MA 01003
aschein@cs.umass.edu

**Mingyuan Zhou**
McCombs School of Business
The University of Texas at Austin
Austin, TX 78712
mingyuan.zhou@mccombs.utexas.edu

**Hanna Wallach**
Microsoft Research New York
641 Avenue of the Americas
New York, NY 10011
hanna@dirichlet.net

## Abstract

We introduce a new dynamical system for sequentially observed multivariate count data. This model is based on the gamma–Poisson construction—a natural choice for count data—and relies on a novel Bayesian nonparametric prior that ties and shrinks the model parameters, thus avoiding overfitting. We present an efficient MCMC inference algorithm that advances recent work on augmentation schemes for inference in negative binomial models. Finally, we demonstrate the model's inductive bias using a variety of real-world data sets, showing that it exhibits superior predictive performance over other models and infers highly interpretable latent structure.

## 1 Introduction

Sequentially observed count vectors $\boldsymbol{y}^{(1)}, \ldots, \boldsymbol{y}^{(T)}$ are the main object of study in many real-world applications, including text analysis, social network analysis, and recommender systems. Count data pose unique statistical and computational challenges when they are high-dimensional, sparse, and overdispersed, as is often the case in real-world applications. For example, when tracking counts of user interactions in a social network, only a tiny fraction of possible edges are ever active, exhibiting bursty periods of activity when they are. Models of such data should exploit this sparsity in order to scale to high dimensions and be robust to overdispersed temporal patterns. In addition to these characteristics, sequentially observed multivariate count data often exhibit complex dependencies within and across time steps. For example, scientific papers about one topic may encourage researchers to write papers about another related topic in the following year. Models of such data should therefore capture the topic structure of individual documents as well as the excitatory relationships between topics.

The linear dynamical system (LDS) is a widely used model for sequentially observed data, with many well-developed inference techniques based on the Kalman filter [1, 2]. The LDS assumes that each sequentially observed $V$-dimensional vector $\boldsymbol{r}^{(t)}$ is real valued and Gaussian distributed: $\boldsymbol{r}^{(t)} \sim \mathcal{N}(\Phi \, \boldsymbol{\theta}^{(t)}, \Sigma)$, where $\boldsymbol{\theta}^{(t)} \in \mathbb{R}^K$ is a latent state, with $K$ components, that is linked to the observed space via $\Phi \in \mathbb{R}^{V \times K}$. The LDS derives its expressive power from the way it assumes that the latent states evolve: $\boldsymbol{\theta}^{(t)} \sim \mathcal{N}(\Pi \, \boldsymbol{\theta}^{(t-1)}, \Delta)$, where $\Pi \in \mathbb{R}^{K \times K}$ is a transition matrix that captures between-component dependencies across time steps. Although the LDS can be linked to non-real observations via the extended Kalman filter [3], it cannot efficiently model real-world count data because inference is $\mathcal{O}((K + V)^3)$ and thus scales poorly with the dimensionality of the data [2].

Many previous approaches to modeling sequentially observed count data rely on the generalized linear modeling framework [4] to link the observations to a latent Gaussian space—e.g., via the Poisson–lognormal link [5]. Researchers have used this construction to factorize sequentially observed count matrices under a Poisson likelihood, while modeling the temporal structure using well-studied Gaussian techniques [6, 7]. Most of these previous approaches assume a simple Gaussian state-space model—i.e., $\boldsymbol{\theta}^{(t)} \sim \mathcal{N}(\boldsymbol{\theta}^{(t-1)}, \Delta)$—that lacks the expressive transition structure of the LDS; one notable exception is the Poisson linear dynamical system [8]. In practice, these approaches exhibit prohibitive computational complexity in high dimensions, and the Gaussian assumption may fail to accommodate the burstiness often inherent to real-world count data [9].

We present the *Poisson–gamma dynamical system (PGDS)*—a new dynamical system, based on the gamma–Poisson construction, that supports the expressive transition structure of the LDS. This model naturally handles overdispersed data. We introduce a new Bayesian nonparametric prior to automatically infer the model's rank. We develop an elegant and efficient algorithm for inferring the parameters of the transition structure that advances recent work on augmentation schemes for inference in negative binomial models [10] and scales with the number of non-zero counts, thus exploiting the sparsity inherent to real-world count data. We examine the way in which the dynamical gamma–Poisson construction propagates information and derive the model's steady state, which involves the Lambert W function [11]. Finally, we use the PGDS to analyze a diverse range of real-world data sets, showing that it exhibits excellent predictive performance on smoothing and forecasting tasks and infers interpretable latent structure, an example of which is depicted in figure 1.

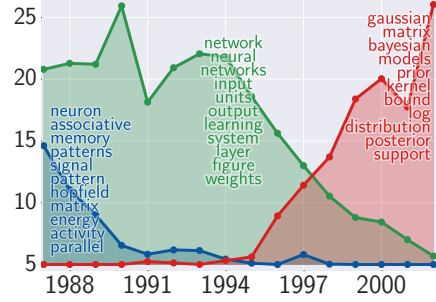

Figure 1: The time-step factors for three components inferred by the PGDS from a corpus of NIPS papers. Each component is associated with a feature factor for each word type in the corpus; we list the words with the largest factors. The inferred structure tells a familiar story about the rise and fall of certain subfields of machine learning.

## 2 Poisson–Gamma Dynamical Systems

We can represent a data set of $V$-dimensional sequentially observed count vectors $\boldsymbol{y}^{(1)}, \ldots, \boldsymbol{y}^{(T)}$ as a $V \times T$ count matrix $Y$. The PGDS models a single count $y_v^{(t)} \in \{0, 1, \ldots\}$ in this matrix as follows:

$$y_v^{(t)} \sim \text{Pois}(\delta^{(t)} \textstyle\sum_{k=1}^{K} \phi_{vk}\, \theta_k^{(t)}) \text{ and } \theta_k^{(t)} \sim \text{Gam}(\tau_0 \textstyle\sum_{k_2=1}^{K} \pi_{kk_2}\, \theta_{k_2}^{(t-1)}, \tau_0), \qquad (1)$$

where the latent factors $\phi_{vk}$ and $\theta_k^{(t)}$ are both positive, and represent the strength of feature $v$ in component $k$ and the strength of component $k$ at time step $t$, respectively. The scaling factor $\delta^{(t)}$ captures the scale of the counts at time step $t$, and therefore obviates the need to rescale the data as a preprocessing step. We refer to the PGDS as *stationary* if $\delta^{(t)} = \delta$ for $t = 1, \ldots, T$. We can view the feature factors as a $V \times K$ matrix $\Phi$ and the time-step factors as a $T \times K$ matrix $\Theta$. Because we can also collectively view the scaling factors and time-step factors as a $T \times K$ matrix $\Psi$, where element $\psi_{tk} = \delta^{(t)} \theta_k^{(t)}$, the PGDS is a form of Poisson matrix factorization: $Y \sim \text{Pois}(\Phi \Psi^T)$ [12, 13, 14, 15].

The PGDS is characterized by its expressive transition structure, which assumes that each time-step factor $\theta_k^{(t)}$ is drawn from a gamma distribution, whose shape parameter is a linear combination of the $K$ factors at the previous time step. The latent transition weights $\pi_{11}, \ldots, \pi_{k_1 k_2}, \ldots, \pi_{KK}$, which we can view as a $K \times K$ transition matrix $\Pi$, capture the excitatory relationships between components. The vector $\boldsymbol{\theta}^{(t)} = (\theta_1^{(t)}, \ldots, \theta_K^{(t)})$ has an expected value of $\mathbb{E}[\boldsymbol{\theta}^{(t)} \,|\, \boldsymbol{\theta}^{(t-1)}, \Pi] = \Pi \, \boldsymbol{\theta}^{(t-1)}$ and is therefore analogous to a latent state in the the LDS. The concentration parameter $\tau_0$ determines the variance of $\boldsymbol{\theta}^{(t)}$—specifically, $\text{Var}\,(\boldsymbol{\theta}^{(t)} \,|\, \boldsymbol{\theta}^{(t-1)}, \Pi) = (\Pi \, \boldsymbol{\theta}^{(t-1)}) \, \tau_0^{-1}$—without affecting its expected value.

To model the strength of each component, we introduce $K$ component weights $\boldsymbol{\nu} = (\nu_1, \ldots, \nu_K)$ and place a shrinkage prior over them. We assume that the time-step factors and transition weights for component $k$ are tied to its component weight $\nu_k$. Specifically, we define the following structure:

$$\theta_k^{(1)} \sim \text{Gam}(\tau_0 \, \nu_k, \tau_0) \text{ and } \boldsymbol{\pi}_k \sim \text{Dir}(\nu_1 \nu_k, \ldots, \xi \nu_k \ldots, \nu_K \nu_k) \text{ and } \nu_k \sim \text{Gam}(\tfrac{\gamma_0}{K}, \beta), \qquad (2)$$

where $\boldsymbol{\pi}_k = (\pi_{1k}, \ldots, \pi_{Kk})$ is the $k^{\text{th}}$ column of $\Pi$. Because $\sum_{k_1=1}^{K} \pi_{k_1 k} = 1$, we can interpret $\pi_{k_1 k}$ as the probability of transitioning from component $k$ to component $k_1$. (We note that interpreting $\Pi$ as a stochastic transition matrix relates the PGDS to the discrete hidden Markov model.) For a fixed value of $\gamma_0$, increasing $K$ will encourage many of the component weights to be small. A small value of $\nu_k$ will shrink $\theta_k^{(1)}$, as well as the transition weights in the $k^{\text{th}}$ row of $\Pi$. Small values of the transition weights in the $k^{\text{th}}$ row of $\Pi$ therefore prevent component $k$ from being excited by the other components and by itself. Specifically, because the shape parameter for the gamma prior over $\theta_k^{(t)}$ involves a linear combination of $\boldsymbol{\theta}^{(t-1)}$ and the transition weights in the $k^{\text{th}}$ row of $\Pi$, small transition weights will result in a small shape parameter, shrinking $\theta_k^{(t)}$. Thus, the component weights play a critical role in the PGDS by enabling it to automatically turn off any unneeded capacity and avoid overfitting.

Finally, we place Dirichlet priors over the feature factors and draw the other parameters from a non-informative gamma prior: $\boldsymbol{\phi}_k = (\phi_{1k}, \ldots, \phi_{Vk}) \sim \text{Dir}(\eta_0, \ldots, \eta_0)$ and $\delta^{(t)}, \xi, \beta \sim \text{Gam}(\epsilon_0, \epsilon_0)$. The PGDS therefore has four positive hyperparameters to be set by the user: $\tau_0$, $\gamma_0$, $\eta_0$, and $\epsilon_0$.

**Bayesian nonparametric interpretation:** As $K \to \infty$, the component weights and their corresponding feature factor vectors constitute a draw $G = \sum_{k=1}^{\infty} \nu_k \mathbb{1}_{\boldsymbol{\phi}_k}$ from a gamma process $\text{GamP}(G_0, \beta)$, where $\beta$ is a scale parameter and $G_0$ is a finite and continuous base measure over a complete separable metric space $\Omega$ [16]. Models based on the gamma process have an inherent shrinkage mechanism because the number of atoms with weights greater than $\varepsilon > 0$ follows a Poisson distribution with a finite mean—specifically, $\text{Pois}(\gamma_0 \int_\varepsilon^\infty d\nu \, \nu^{-1} \exp(-\beta \nu))$, where $\gamma_0 = G_0(\Omega)$ is the total mass under the base measure. This interpretation enables us to view the priors over $\Pi$ and $\Theta$ as novel stochastic processes, which we call the *column-normalized relational gamma process* and the *recurrent gamma process*, respectively. We provide the definitions of these processes in the supplementary material.

**Non-count observations:** The PGDS can also model non-count data by linking the observed vectors to latent counts. A binary observation $b_v^{(t)}$ can be linked to a latent Poisson count $y_v^{(t)}$ via the Bernoulli–Poisson distribution: $b_v^{(t)} = \mathbb{1}(y_v^{(t)} \geq 1)$ and $y_v^{(t)} \sim \text{Pois}(\delta^{(t)} \sum_{k=1}^{K} \phi_{vk} \theta_k^{(t)})$ [17]. Similarly, a real-valued observation $r_v^{(t)}$ can be linked to a latent Poisson count $y_v^{(t)}$ via the Poisson randomized gamma distribution [18]. Finally, Basbug and Engelhardt [19] recently showed that many types of non-count matrices can be linked to a latent count matrix via the compound Poisson distribution [20].

# 3 MCMC Inference

MCMC inference for the PGDS consists of drawing samples of the model parameters from their joint posterior distribution given an observed count matrix $Y$ and the model hyperparameters $\tau_0$, $\gamma_0$, $\eta_0$, $\epsilon_0$. In this section, we present a Gibbs sampling algorithm for drawing these samples. At a high level, our approach is similar to that used to develop Gibbs sampling algorithms for several other related models [10, 21, 22, 17]; however, we extend this approach to handle the unique properties of the PGDS. The main technical challenge is sampling $\Theta$ from its conditional posterior, which does not have a closed form. We address this challenge by introducing a set of auxiliary variables. Under this augmented version of the model, marginalizing over $\Theta$ becomes tractable and its conditional posterior has a closed form. Moreover, by introducing these auxiliary variables and marginalizing over $\Theta$, we obtain an alternative model specification that we can subsequently exploit to obtain closed-form conditional posteriors for $\Pi$, $\boldsymbol{\nu}$, and $\xi$. We marginalize over $\Theta$ by performing a "backward filtering" pass, starting with $\boldsymbol{\theta}^{(T)}$. We repeatedly exploit the following three definitions in order to do this.

*Definition 1:* If $y_\cdot = \sum_{n=1}^{N} y_n$, where $y_n \sim \text{Pois}(\theta_n)$ are independent Poisson-distributed random variables, then $(y_1, \ldots, y_N) \sim \text{Mult}(y_\cdot, (\frac{\theta_1}{\sum_{n=1}^{N} \theta_n}, \ldots, \frac{\theta_N}{\sum_{n=1}^{N} \theta_n}))$ and $y_\cdot \sim \text{Pois}(\sum_{n=1}^{N} \theta_n)$ [23, 24].

*Definition 2:* If $y \sim \text{Pois}(c\theta)$, where $c$ is a constant, and $\theta \sim \text{Gam}(a, b)$, then $y \sim \text{NB}(a, \frac{c}{b+c})$ is a negative binomial–distributed random variable. We can equivalently parameterize it as $y \sim \text{NB}(a, g(\zeta))$, where $g(z) = 1 - \exp(-z)$ is the Bernoulli–Poisson link [17] and $\zeta = \ln(1 + \frac{c}{b})$.

*Definition 3:* If $y \sim \text{NB}(a, g(\zeta))$ and $l \sim \text{CRT}(y, a)$ is a Chinese restaurant table–distributed random variable, then $y$ and $l$ are equivalently jointly distributed as $y \sim \text{SumLog}(l, g(\zeta))$ and $l \sim \text{Pois}(a\zeta)$ [21]. The sum logarithmic distribution is further defined as the sum of $l$ independent and identically logarithmic-distributed random variables—i.e., $y = \sum_{i=1}^{l} x_i$ and $x_i \sim \text{Log}(g(\zeta))$.

**Marginalizing over $\Theta$:** We first note that we can re-express the Poisson likelihood in equation 1 in terms of latent subcounts [13]: $y_v^{(t)} = y_{v\cdot}^{(t)} = \sum_{k=1}^K y_{vk}^{(t)}$ and $y_{vk}^{(t)} \sim \text{Pois}(\delta^{(t)} \phi_{vk} \theta_k^{(t)})$. We then define $y_{\cdot k}^{(t)} = \sum_{v=1}^V y_{vk}^{(t)}$. Via definition 1, we obtain $y_{\cdot k}^{(t)} \sim \text{Pois}(\delta^{(t)} \theta_k^{(t)})$ because $\sum_{v=1}^V \phi_{vk} = 1$.

We start with $\theta_k^{(T)}$ because none of the other time-step factors depend on it in their priors. Via definition 2, we can immediately marginalize over $\theta_k^{(T)}$ to obtain the following equation:

$$y_{\cdot k}^{(T)} \sim \text{NB}(\tau_0 \sum_{k_2=1}^K \pi_{kk_2} \theta_{k_2}^{(T-1)}, g(\zeta^{(T)})), \text{ where } \zeta^{(T)} = \ln\left(1 + \tfrac{\delta^{(T)}}{\tau_0}\right). \tag{3}$$

Next, we marginalize over $\theta_k^{(T-1)}$. To do this, we introduce an auxiliary variable: $l_k^{(T)} \sim \text{CRT}(y_{\cdot k}^{(T)}, \tau_0 \sum_{k_2=1}^K \pi_{kk_2} \theta_{k_2}^{(T-1)})$. We can then re-express the joint distribution over $y_{\cdot k}^{(T)}$ and $l_k^{(T)}$ as

$$y_{\cdot k}^{(T)} \sim \text{SumLog}(l_k^{(T)}, g(\zeta^{(T)}) \text{ and } l_k^{(T)} \sim \text{Pois}(\zeta^{(T)} \tau_0 \sum_{k_2=1}^K \pi_{kk_2} \theta_{k_2}^{(T-1)}). \tag{4}$$

We are still unable to marginalize over $\theta_k^{(T-1)}$ because it appears in a sum in the parameter of the Poisson distribution over $l_k^{(T)}$; however, via definition 1, we can re-express this distribution as

$$l_k^{(T)} = l_{k\cdot}^{(T)} = \sum_{k_2=1}^K l_{kk_2}^{(T)} \text{ and } l_{kk_2}^{(T)} \sim \text{Pois}(\zeta^{(T)} \tau_0 \pi_{kk_2} \theta_{k_2}^{(T-1)}). \tag{5}$$

We then define $l_{\cdot k}^{(T)} = \sum_{k_1=1}^K l_{k_1 k}^{(T)}$. Again via definition 1, we can express the distribution over $l_{\cdot k}^{(T)}$ as $l_{\cdot k}^{(T)} \sim \text{Pois}(\zeta^{(T)} \tau_0 \theta_k^{(T-1)})$. We note that this expression does not depend on the transition weights because $\sum_{k_1=1}^K \pi_{k_1 k} = 1$. We also note that definition 1 implies that $(l_{1k}^{(T)}, \ldots, l_{Kk}^{(T)}) \sim \text{Mult}(l_{\cdot k}^{(T)}, (\pi_1, \ldots, \pi_K))$. Next, we introduce $m_k^{(T-1)} = y_{\cdot k}^{(T-1)} + l_{\cdot k}^{(T)}$, which summarizes all of the information about the data at time steps $T-1$ and $T$ via $y_{\cdot k}^{(T-1)}$ and $l_{\cdot k}^{(T)}$, respectively. Because $y_{\cdot k}^{(T-1)}$ and $l_{\cdot k}^{(T)}$ are both Poisson distributed, we can use definition 1 to obtain

$$m_k^{(T-1)} \sim \text{Pois}(\theta_k^{(T-1)}(\delta^{(T-1)} + \zeta^{(T)} \tau_0)). \tag{6}$$

Combining this likelihood with the gamma prior in equation 1, we can marginalize over $\theta_k^{(T-1)}$:

$$m_k^{(T-1)} \sim \text{NB}(\tau_0 \sum_{k_2=1}^K \pi_{kk_2} \theta_{k_2}^{(T-2)}, g(\zeta^{(T-1)})), \text{ where } \zeta^{(T-1)} = \ln\left(1 + \tfrac{\delta^{(T-1)}}{\tau_0} + \zeta^{(T)}\right). \tag{7}$$

We then introduce $l_k^{(T-1)} \sim \text{CRT}(m_k^{(T-1)}, \tau_0 \sum_{k_2=1}^K \pi_{kk_2} \theta_{k_2}^{(T-2)})$ and re-express the joint distribution over $l_k^{(T-1)}$ and $m_k^{(T-1)}$ as the product of a Poisson and a sum logarithmic distribution, similar to equation 4. This then allows us to marginalize over $\theta_k^{(T-2)}$ to obtain a negative binomial distribution. We can repeat the same process all the way back to $t = 1$, where marginalizing over $\theta_k^{(1)}$ yields $m_k^{(1)} \sim \text{NB}(\tau_0 \nu_k, g(\zeta^{(1)}))$. We note that just as $m_k^{(t)}$ summarizes all of the information about the data at time steps $t, \ldots, T$, $\zeta^{(t)} = \ln\left(1 + \tfrac{\delta^{(t)}}{\tau_0} + \zeta^{(t+1)}\right)$ summarizes all of the information about $\delta^{(t)}, \ldots, \delta^{(T)}$.

As we mentioned previously, introducing these auxiliary variables and marginalizing over $\Theta$ also enables us to define an alternative model specification that we can exploit to obtain closed-form conditional posteriors for $\Pi$, $\boldsymbol{\nu}$, and $\xi$. We provide part of its generative process in figure 2. We define $m_k^{(T)} = y_{\cdot k}^{(T)} + l_{\cdot k}^{(T+1)}$, where $l_{\cdot k}^{(T+1)} = 0$, and $\zeta^{(T+1)} = 0$ so that we can present the alternative model specification concisely.

$l_{k\cdot}^{(1)} \sim \text{Pois}(\zeta^{(1)} \tau_0 \nu_k)$
$(l_{1k}^{(t)}, \ldots, l_{Kk}^{(t)}) \sim \text{Mult}(l_{\cdot k}^{(t)}, (\pi_{1k}, \ldots, \pi_{Kk}))$ for $t > 1$
$l_{k\cdot}^{(t)} = \sum_{k_2=1}^K l_{kk_2}^{(t)}$ for $t > 1$
$m_k^{(t)} \sim \text{SumLog}(l_{k\cdot}^{(t)}, g(\zeta^{(t)}))$
$(y_{\cdot k}^{(t)}, l_{\cdot k}^{(t+1)}) \sim \text{Bin}(m_k^{(t)}, (\tfrac{\delta^{(t)}}{\delta^{(t)} + \zeta^{(t+1)} \tau_0}, \tfrac{\zeta^{(t+1)} \tau_0}{\delta^{(t)} + \zeta^{(t+1)} \tau_0}))$
$(y_{1k}^{(t)}, \ldots, y_{Vk}^{(t)}) \sim \text{Mult}(y_{\cdot k}^{(t)}, (\phi_{1k}, \ldots, \phi_{Vk}))$

Figure 2: Alternative model specification.

**Steady state:** We draw particular attention to the backward pass $\zeta^{(t)} = \ln\left(1 + \tfrac{\delta^{(t)}}{\tau_0} + \zeta^{(t+1)}\right)$ that propagates information about $\delta^{(t)}, \ldots, \delta^{(T)}$ as we marginalize over $\Theta$. In the case of the stationary PGDS—i.e., $\delta^{(t)} = \delta$—the backward pass has a fixed point that we define in the following proposition.

*Proposition 1:* The backward pass has a fixed point of $\zeta^{\star} = -\mathbb{W}_{-1}(-\exp(-1-\frac{\delta}{\tau_0})) - 1 - \frac{\delta}{\tau_0}$.

The function $\mathbb{W}_{-1}(\cdot)$ is the lower real part of the Lambert W function [11]. We prove this proposition in the supplementary material. During inference, we perform the $\mathcal{O}(T)$ backward pass repeatedly. The existence of a fixed point means that we can assume the stationary PGDS is in its steady state and replace the backward pass with an $\mathcal{O}(1)$ computation[1] of the fixed point $\zeta^*$. To make this assumption, we must also assume that $l_{\cdot k}^{(T+1)} \sim \text{Pois}(\zeta^{\star}\tau_0\theta_k^{(T)})$ instead of $l_{\cdot k}^{(T+1)} = 0$. We note that an analogous steady-state approximation exists for the LDS and is routinely exploited to reduce computation [25].

**Gibbs sampling algorithm:** Given $Y$ and the hyperparameters, Gibbs sampling involves resampling each auxiliary variable or model parameter from its conditional posterior. Our algorithm involves a "backward filtering" pass and a "forward sampling" pass, which together form a "backward filtering–forward sampling" algorithm. We use $- \setminus \Theta^{(\geq t)}$ to denote everything excluding $\boldsymbol{\theta}^{(t)}, \ldots, \boldsymbol{\theta}^{(T)}$.

*Sampling the auxiliary variables:* This step is the "backward filtering" pass. For the stationary PGDS in its steady state, we first compute $\zeta^*$ and draw $(l_{\cdot k}^{(T+1)} \,|\, -) \sim \text{Pois}(\zeta^{\star}\tau_0\theta_k^{(T)})$. For the other variants of the model, we set $l_{\cdot k}^{(T+1)} = \zeta^{(T+1)} = 0$. Then, working backward from $t = T, \ldots, 2$, we draw

$$(l_{k\cdot}^{(t)} \,|\, - \setminus \Theta^{(\geq t)}) \sim \text{CRT}(y_{\cdot k}^{(t)} + l_{\cdot k}^{(t+1)}, \tau_0\textstyle\sum_{k_2=1}^{K}\pi_{kk_2}\theta_{k_2}^{(t-1)}) \text{ and} \tag{8}$$

$$(l_{k1}^{(t)}, \ldots, l_{kK}^{(t)} \,|\, - \setminus \Theta^{(\geq t)}) \sim \text{Mult}(l_{k\cdot}^{(t)}, (\tfrac{\pi_{k1}\theta_1^{(t-1)}}{\sum_{k_2=1}^{K}\pi_{kk_2}\theta_{k_2}^{(t-1)}}, \ldots, \tfrac{\pi_{kK}\theta_K^{(t-1)}}{\sum_{k_2=1}^{K}\pi_{kk_2}\theta_{k_2}^{(t-1)}})). \tag{9}$$

After using equations 8 and 9 for all $k = 1, \ldots, K$, we then set $l_{\cdot k}^{(t)} = \sum_{k_1=1}^{K}l_{k_1 k}^{(t)}$. For the non-steady-state variants, we also set $\zeta^{(t)} = \ln(1 + \frac{\delta^{(t)}}{\tau_0} + \zeta^{(t+1)})$; for the steady-state variant, we set $\zeta^{(t)} = \zeta^*$.

*Sampling $\Theta$:* We sample $\Theta$ from its conditional posterior by performing a "forward sampling" pass, starting with $\boldsymbol{\theta}^{(1)}$. Conditioned on the values of $l_{\cdot k}^{(2)}, \ldots, l_{\cdot k}^{(T+1)}$ and $\zeta^{(2)}, \ldots, \zeta^{(T+1)}$ obtained via the "backward filtering" pass, we sample forward from $t = 1, \ldots, T$, using the following equations:

$$(\theta_k^{(1)} \,|\, - \setminus \Theta) \sim \text{Gam}(y_{\cdot k}^{(1)} + l_{\cdot k}^{(2)} + \tau_0\nu_k, \tau_0 + \delta^{(1)} + \zeta^{(2)}\tau_0) \text{ and} \tag{10}$$

$$(\theta_k^{(t)} \,|\, - \setminus \Theta^{(\geq t)}) \sim \text{Gam}(y_{\cdot k}^{(t)} + l_{\cdot k}^{(t+1)} + \tau_0\textstyle\sum_{k_2=1}^{K}\pi_{kk_2}\theta_{k_2}^{(t-1)}, \tau_0 + \delta^{(t)} + \zeta^{(t+1)}\tau_0). \tag{11}$$

*Sampling $\Pi$:* The alternative model specification, with $\Theta$ marginalized out, assumes that $(l_{1k}^{(t)}, \ldots, l_{Kk}^{(t)}) \sim \text{Mult}(l_{\cdot k}^{(t)}, (\pi_{1k}, \ldots, \pi_{Kk}))$. Therefore, via Dirichlet–multinomial conjugacy,

$$(\boldsymbol{\pi}_k \,|\, - \setminus \Theta) \sim \text{Dir}(\nu_1\nu_k + \textstyle\sum_{t=1}^{T}l_{1k}^{(t)}, \ldots, \xi\nu_k + \sum_{t=1}^{T}l_{kk}^{(t)}, \ldots, \nu_K\nu_k + \sum_{t=1}^{T}l_{Kk}^{(t)}). \tag{12}$$

*Sampling $\boldsymbol{\nu}$ and $\xi$:* We use the alternative model specification to obtain closed-form conditional posteriors for $\nu_k$ and $\xi$. First, we marginalize over $\boldsymbol{\pi}_k$ to obtain a Dirichlet–multinomial distribution. When augmented with a beta-distributed auxiliary variable, the Dirichlet–multinomial distribution is proportional to the negative binomial distribution [26]. We draw such an auxiliary variable, which we use, along with negative binomial augmentation schemes, to derive closed-form conditional posteriors for $\nu_k$ and $\xi$. We provide these posteriors, along with their derivations, in the supplementary material.

We also provide the conditional posteriors for the remaining model parameters—$\Phi$, $\delta^{(1)}, \ldots, \delta^{(T)}$, and $\beta$—which we obtain via Dirichlet–multinomial, gamma–Poisson, and gamma–gamma conjugacy.

## 4 Experiments

In this section, we compare the predictive performance of the PGDS to that of the LDS and that of gamma process dynamic Poisson factor analysis (GP-DPFA) [22]. GP-DPFA models a single count in $Y$ as $y_v^{(t)} \sim \text{Pois}(\sum_{k=1}^{K}\lambda_k\phi_{vk}\theta_k^{(t)})$, where each component's time-step factors evolve as a simple gamma Markov chain, independently of those belonging to the other components: $\theta_k^{(t)} \sim \text{Gam}(\theta_k^{(t-1)}, c^{(t)})$. We consider the stationary variants of all three models.[2] We used five data sets, and tested each model on two time-series prediction tasks: smoothing—i.e., predicting $y_v^{(t)}$ given

$y_v^{(1)}, \ldots, y_v^{(t-1)}, y_v^{(t+1)}, \ldots, y_v^{(T)}$—and forecasting—i.e., predicting $y_v^{(T+s)}$ given $y_v^{(1)}, \ldots, y_v^{(T)}$ for some $s \in \{1, 2, \ldots\}$ [27]. We provide brief descriptions of the data sets below before reporting results.

*Global Database of Events, Language, and Tone (GDELT):* GDELT is an international relations data set consisting of country-to-country interaction events of the form "country $i$ took action $a$ toward country $j$ at time $t$," extracted from news corpora. We created five count matrices, one for each year from 2001 through 2005. We treated directed pairs of countries $i \rightarrow j$ as features and counted the number of events for each pair during each day. We discarded all pairs with fewer than twenty-five total events, leaving $T = 365$, around $V \approx 9,000$, and three to six million events for each matrix.

*Integrated Crisis Early Warning System (ICEWS):* ICEWS is another international relations event data set extracted from news corpora. It is more highly curated than GDELT and contains fewer events. We therefore treated undirected pairs of countries $i \leftrightarrow j$ as features. We created three count matrices, one for 2001–2003, one for 2004–2006, and one for 2007–2009. We counted the number of events for each pair during each three-day time step, and again discarded all pairs with fewer than twenty-five total events, leaving $T = 365$, around $V \approx 3,000$, and 1.3 to 1.5 million events for each matrix.

*State-of-the-Union transcripts (SOTU):* The SOTU corpus contains the text of the annual SOTU speech transcripts from 1790 through 2014. We created a single count matrix with one column per year. After discarding stopwords, we were left with $T = 225$, $V = 7,518$, and 656,949 tokens.

*DBLP conference abstracts (DBLP):* DBLP is a database of computer science research papers. We used the subset of this corpus that Acharya et al. used to evaluate GP-DPFA [22]. This subset corresponds to a count matrix with $T = 14$ columns, $V = 1,771$ unique word types, and 13,431 tokens.

*NIPS corpus (NIPS):* The NIPS corpus contains the text of every NIPS conference paper from 1987 to 2003. We created a single count matrix with one column per year. We treated unique word types as features and discarded all stopwords, leaving $T = 17$, $V = 9,836$, and 3.1 million tokens.

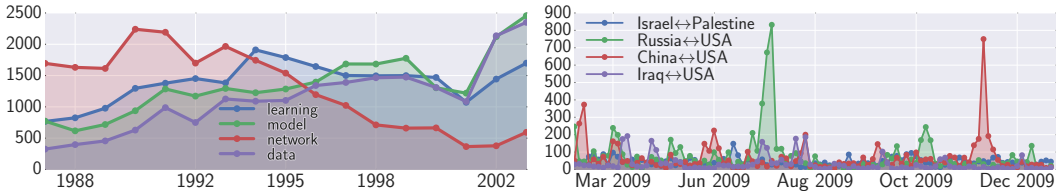

Figure 3: $y_v^{(t)}$ over time for the top four features in the NIPS (left) and ICEWS (right) data sets.

**Experimental design:** For each matrix, we created four masks indicating some randomly selected subset of columns to treat as held-out data. For the event count matrices, we held out six (non-contiguous) time steps between $t = 2$ and $t = T - 3$ to test the models' smoothing performance, as well as the last two time steps to test their forecasting performance. The other matrices have fewer time steps. For the SOTU matrix, we therefore held out five time steps between $t = 2$ and $t = T - 2$, as well as $t = T$. For the NIPS and DBLP matrices, which contain substantially fewer time steps than the SOTU matrix, we held out three time steps between $t = 2$ and $t = T - 2$, as well as $t = T$.

For each matrix, mask, and model combination, we ran inference four times.[3] For the PGDS and GP-DPFA, we performed 6,000 Gibbs sampling iterations, imputing the missing counts from the "smoothing" columns at the same time as sampling the model parameters. We then discarded the first 4,000 samples and retained every hundredth sample thereafter. We used each of these samples to predict the missing counts from the "forecasting" columns. We then averaged the predictions over the samples. For the LDS, we ran EM to learn the model parameters. Then, given these parameter values, we used the Kalman filter and smoother [1] to predict the held-out data. In practice, for all five data sets, $V$ was too large for us to run inference for the LDS, which is $\mathcal{O}((K + V)^3)$ [2], using all $V$ features. We therefore report results from two independent sets of experiments: one comparing all three models using only the top $V = 1,000$ features for each data set, and one comparing the PGDS to just GP-DPFA using all the features. The first set of experiments is generous to the LDS because the Poisson distribution is well approximated by the Gaussian distribution when its mean is large.

Table 1: Results for the smoothing ("S") and forecasting ("F") tasks. For both error measures, lower values are better. We also report the number of time steps $T$ and the burstiness $\hat{B}$ of each data set.

| | $T$ | $\hat{B}$ | Task | Mean Relative Error (MRE) | | | Mean Absolute Error (MAE) | | |
|---|---|---|---|---|---|---|---|---|---|
| | | | | PGDS | GP-DPFA | LDS | PGDS | GP-DPFA | LDS |
| GDELT | 365 | 1.27 | S | **2.335** $_{\pm0.19}$ | 2.951 $_{\pm0.32}$ | 3.493 $_{\pm0.53}$ | 9.366 $_{\pm2.19}$ | **9.278** $_{\pm2.01}$ | 10.098 $_{\pm2.39}$ |
| | | | F | **2.173** $_{\pm0.41}$ | 2.207 $_{\pm0.42}$ | 2.397 $_{\pm0.29}$ | **7.002** $_{\pm1.43}$ | 7.095 $_{\pm1.67}$ | 7.047 $_{\pm1.25}$ |
| ICEWS | 365 | 1.10 | S | **0.808** $_{\pm0.11}$ | 0.877 $_{\pm0.12}$ | 1.023 $_{\pm0.15}$ | **2.867** $_{\pm0.56}$ | 2.872 $_{\pm0.56}$ | 3.104 $_{\pm0.60}$ |
| | | | F | **0.743** $_{\pm0.17}$ | 0.792 $_{\pm0.17}$ | 0.937 $_{\pm0.31}$ | **1.788** $_{\pm0.47}$ | 1.894 $_{\pm0.50}$ | 1.973 $_{\pm0.62}$ |
| SOTU | 225 | 1.45 | S | **0.233** $_{\pm0.01}$ | 0.238 $_{\pm0.01}$ | 0.260 $_{\pm0.01}$ | **0.408** $_{\pm0.01}$ | 0.414 $_{\pm0.01}$ | 0.448 $_{\pm0.00}$ |
| | | | F | **0.171** $_{\pm0.00}$ | 0.173 $_{\pm0.00}$ | 0.225 $_{\pm0.01}$ | 0.323 $_{\pm0.00}$ | **0.314** $_{\pm0.00}$ | 0.370 $_{\pm0.00}$ |
| DBLP | 14 | 1.64 | S | 0.417 $_{\pm0.03}$ | 0.422 $_{\pm0.05}$ | **0.405** $_{\pm0.05}$ | **0.771** $_{\pm0.03}$ | 0.782 $_{\pm0.06}$ | 0.831 $_{\pm0.01}$ |
| | | | F | **0.322** $_{\pm0.00}$ | 0.323 $_{\pm0.00}$ | 0.369 $_{\pm0.06}$ | 0.747 $_{\pm0.01}$ | **0.715** $_{\pm0.00}$ | 0.943 $_{\pm0.07}$ |
| NIPS | 17 | 0.33 | S | 0.415 $_{\pm0.07}$ | **0.392** $_{\pm0.07}$ | 1.609 $_{\pm0.43}$ | 29.940 $_{\pm2.95}$ | **28.138** $_{\pm3.08}$ | 108.378 $_{\pm15.44}$ |
| | | | F | 0.343 $_{\pm0.01}$ | **0.312** $_{\pm0.00}$ | 0.642 $_{\pm0.14}$ | 62.839 $_{\pm0.37}$ | **52.963** $_{\pm0.52}$ | 95.495 $_{\pm10.52}$ |

**Results:** We used two error measures—mean relative error (MRE) and mean absolute error (MAE)—to compute the models' smoothing and forecasting scores for each matrix and mask combination. We then averaged these scores over the masks. For the data sets with multiple matrices, we also averaged the scores over the matrices. The two error measures differ as follows: MRE accommodates the scale of the data, while MAE does not. This is because relative error—which we define as $\frac{|y_v^{(t)} - \hat{y}_v^{(t)}|}{1 + y_v^{(t)}}$, where $y_v^{(t)}$ is the true count and $\hat{y}_v^{(t)}$ is the prediction—divides the absolute error by the true count and thus penalizes overpredictions more harshly than underpredictions. MRE is therefore an especially natural choice for data sets that are bursty—i.e., data sets that exhibit short periods of activity that far exceed their mean. Models that are robust to these kinds of overdispersed temporal patterns are less likely to make overpredictions following a burst, and are therefore rewarded accordingly by MRE.

In table 1, we report the MRE and MAE scores for the experiments using the top $V = 1,000$ features. We also report the average burstiness of each data set. We define the burstiness of feature $v$ in matrix $Y$ to be $\hat{B}_v = \frac{1}{T-1} \sum_{t=1}^{T-1} \frac{|y_v^{(t+1)} - y_v^{(t)}|}{\hat{\mu}_v}$, where $\hat{\mu}_v = \frac{1}{T} \sum_{t=1}^{T} y_v^{(t)}$. For each data set, we calculated the burstiness of each feature in each matrix, and then averaged these values to obtain an average burstiness score $\hat{B}$. The PGDS outperformed the LDS and GP-DPFA on seven of the ten prediction tasks when we used MRE to measure the models' performance; when we used MAE, the PGDS outperformed the other models on five of the tasks. In the supplementary material, we also report the results for the experiments comparing the PGDS to GP-DPFA using all the features. The superiority of the PGDS over GP-DPFA is even more pronounced in these results. We hypothesize that the difference between these models is related to the burstiness of the data. For both error measures, the only data set for which GP-DPFA outperformed the PGDS on both tasks was the NIPS data set. This data set has a substantially lower average burstiness score than the other data sets. We provide visual evidence in figure 3, where we display $y_v^{(t)}$ over time for the top four features in the NIPS and ICEWS data sets. For the former, the features evolve smoothly; for the latter, they exhibit bursts of activity.

**Exploratory analysis:** We also explored the latent structure inferred by the PGDS. Because its parameters are positive, they are easy to interpret. In figure 1, we depict three components inferred from the NIPS data set. By examining the time-step factors and feature factors for these components, we see that they capture the decline of research on neural networks between 1987 and 2003, as well as the rise of Bayesian methods in machine learning. These patterns match our prior knowledge.

In figure 4, we depict the three components with the largest component weights inferred by the PGDS from the 2003 GDELT matrix. The top component is in blue, the second is in green, and the third is in red. For each component, we also list the sixteen features (directed pairs of countries) with the largest feature factors. The top component (blue) is most active in March and April, 2003. Its features involve USA, Iraq (IRQ), Great Britain (GBR), Turkey (TUR), and Iran (IRN), among others. This component corresponds to the 2003 invasion of Iraq. The second component (green) exhibits a noticeable increase in activity immediately after April, 2003. Its top features involve Israel (ISR), Palestine (PSE), USA, and Afghanistan (AFG). The third component exhibits a large burst of activity

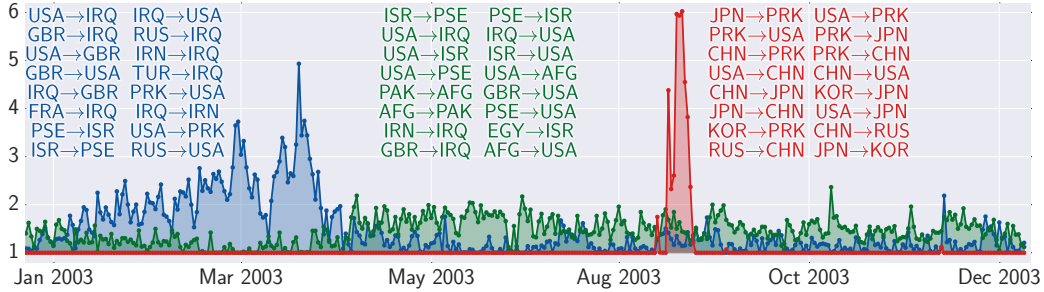

Figure 4: The time-step factors for the top three components inferred by the PGDS from the 2003 GDELT matrix. The top component is in blue, the second is in green, and the third is in red. For each component, we also list the features (directed pairs of countries) with the largest feature factors.

in August, 2003, but is otherwise inactive. Its top features involve North Korea (PRK), South Korea (KOR), Japan (JPN), China (CHN), Russia (RUS), and USA. This component corresponds to the six-party talks—a series of negotiations between these six countries for the purpose of dismantling North Korea's nuclear program. The first round of talks occurred during August 27–29, 2003.

In figure 5, we also show the component weights for the top ten components, along with the corresponding subset of the transition matrix $\Pi$. There are two components with weights greater than one: the components that are depicted in blue and green in figure 4. The transition weights in the corresponding rows of $\Pi$ are also large, meaning that other components are likely to transition to them. As we mentioned previously, the GDELT data set was extracted from news corpora. Therefore, patterns in the data primarily reflect patterns in media coverage of international affairs. We therefore interpret the latent structure inferred by the PGDS in the following way: in 2003, the media briefly covered various major events, including the six-party talks, before quickly returning to a backdrop of the ongoing Iraq war and Israeli–Palestinian relations. By inferring the kind of transition structure depicted in figure 5, the PGDS is able to model persistent, long-term temporal patterns while accommodating the burstiness often inherent to real-world count data. This ability is what enables the PGDS to achieve superior predictive performance over the LDS and GP-DPFA.

## 5 Summary

We introduced the Poisson–gamma dynamical system (PGDS)—a new Bayesian nonparametric model for sequentially observed multivariate count data. This model supports the expressive transition structure of the linear dynamical system, and naturally handles overdispersed data. We presented a novel MCMC inference algorithm that remains efficient for high-dimensional data sets, advancing recent work on augmentation schemes for inference in negative binomial models. Finally, we used the PGDS to analyze five real-world data sets, demonstrating that it exhibits superior smoothing and forecasting performance over two baseline models and infers highly interpretable latent structure.

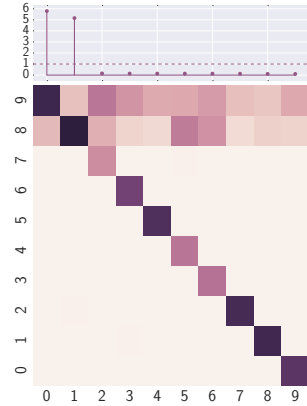

Figure 5: The latent transition structure inferred by the PGDS from the 2003 GDELT matrix. Top: The component weights for the top ten components, in decreasing order from left to right; two of the weights are greater than one. Bottom: The transition weights in the corresponding subset of the transition matrix. This structure means that all components are likely to transition to the top two components.

**Acknowledgments**

We thank David Belanger, Roy Adams, Kostis Gourgoulias, Ben Marlin, Dan Sheldon, and Tim Vieira for many helpful conversations. This work was supported in part by the UMass Amherst CIIR and in part by NSF grants SBE-0965436 and IIS-1320219. Any opinions, findings, conclusions, or recommendations are those of the authors and do not necessarily reflect those of the sponsors.

## Footnotes

[1] Several software packages contain fast implementations of the Lambert W function.

[2] We used the `pykalman` Python library for the LDS and implemented GP-DPFA ourselves.

[3]For the PGDS and GP-DPFA we used $K = 100$. For the PGDS, we set $\tau_0 = 1$, $\gamma_0 = 50$, $\eta_0 = \epsilon_0 = 0.1$. We set the hyperparameters of GP-DPFA to the values used by Acharya et al. [22]. For the LDS, we used the default hyperparameters for `pykalman`, and report results for the best-performing value of $K \in \{5, 10, 25, 50\}$.

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
