[Supplementary Material]

# Supplementary Material for "Poisson–Gamma Dynamical Systems"

**Aaron Schein**
College of Information and Computer Sciences
University of Massachusetts Amherst
Amherst, MA 01003
aschein@cs.umass.edu

**Mingyuan Zhou**
McCombs School of Business
The University of Texas at Austin
Austin, TX 78712
mingyuan.zhou@mccombs.utexas.edu

**Hanna Wallach**
Microsoft Research New York
641 Avenue of the Americas
New York, NY 10011
hanna@dirichlet.net

## 1 Bayesian Nonparametric Interpretation

As $K \to \infty$, the component weights and their corresponding feature factor vectors constitute a draw $G = \sum_{k=1}^{\infty} \nu_k \mathbb{1}_{\phi_k}$ from a gamma process $\mathrm{GamP}\,(G_0, \beta)$, where $\beta$ is a scale parameter and $G_0$ is a finite and continuous base measure over a complete separable metric space $\Omega$ [1]. Models based on the gamma process have an inherent shrinkage mechanism because the number of atoms with weights greater than $\varepsilon > 0$ follows a Poisson distribution with a finite mean—specifically, $\mathrm{Pois}(\gamma_0 \int_{\varepsilon}^{\infty} \mathrm{d}\nu \, \nu^{-1} \exp(-\beta\,\nu))$, where $\gamma_0 = G_0(\Omega)$ is the total mass under the base measure.

This interpretation enables us to view the priors over $\Pi$ and $\Theta$ as a novel stochastic processes. Because of the relationship between the Dirichlet and gamma distributions [1], we can equivalently express the prior over the $k^{\text{th}}$ column of $\Pi$ as $\pi_{k_1 k} = \frac{\lambda_{k_1 k}}{\sum_{k_2=1}^{K} \lambda_{k_1 k_2}}$ for $k_1 = 1, \ldots, K$, where $\lambda_{kk} \sim \mathrm{Gam}(\xi\,\nu_k, c)$, and $\lambda_{k_1 k} \sim \mathrm{Gam}(\nu_{k_1}\nu_k, c)$ for $k_1 \neq k$ are auxiliary variables. As $K \to \infty$, $\sum_{k_1=1}^{\infty} \sum_{k_2=1}^{\infty} \lambda_{k_1 k_2} \mathbb{1}_{(\phi_{k_1}, \phi_{k_2})}$ is a draw from a relational gamma process [2] and $\sum_{k_1=1}^{\infty} \sum_{k_2=1}^{\infty} \pi_{k_1 k_2} \mathbb{1}_{(\phi_{k_1}, \phi_{k_2})}$ is a draw from a *column-normalized relational gamma process*.

Given $\tau_0$, $\Pi$, and $G = \sum_{k=1}^{\infty} \nu_k \mathbb{1}_{\phi_k}$, the prior over $\Theta$ is a *recurrent gamma process*—a sequence of gamma processes, each defined over the product space $\mathbb{R}_+ \times \Omega$—defined recursively as follows:

$$G^{(t)} = \sum_{k=1}^{\infty} \theta_k^{(t)} \mathbb{1}_{\phi_k} \sim \mathrm{GamP}(H^{(t)}, \tau_0) \text{ and } H^{(t)} = \sum_{k=1}^{\infty} (\tau_0 \sum_{k_2=1}^{\infty} \pi_{kk_2}\, \theta_{k_2}^{(t-1)}) \mathbb{1}_{\phi_k}, \quad (1)$$

where $H^{(1)} = \tau_0\, G$. This recursive sequence of gamma processes will never produce a draw with infinite mass at some time step $t$, which would make $H^{(t+1)}$ infinite and violate the entire definition.

*Theorem 1:* The expected sum of $\boldsymbol{\theta}^{(t)}$ is finite and equal to $\mathbb{E}[\sum_{k=1}^{\infty} \theta_k^{(t)}] = \frac{\gamma_0}{\beta}$.

By linearity of expectation, the definition of the recurrent gamma process, and $\sum_{k_1=1}^{\infty} \pi_{k_1 k} = 1$,

$$\mathbb{E}[\sum_{k=1}^{\infty} \theta_k^{(t)}] = \sum_{k=1}^{\infty} \mathbb{E}[\theta_k^{(t)}] = \sum_{k=1}^{\infty} \mathbb{E}[\sum_{k_2=1}^{\infty} \pi_{kk_2}\, \theta_{k_2}^{(t-1)}] = \mathbb{E}[\sum_{k=1}^{\infty} \theta_{k=1}^{(t-1)}]. \quad (2)$$

Then, by induction and the definition of the recurrent gamma process,

$$\mathbb{E}[\sum_{k=1}^{\infty} \theta_k^{(t-1)}] = \ldots = \mathbb{E}[\sum_{k=1}^{\infty} \theta_k^{(1)}] = \frac{\gamma_0}{\beta}. \quad (3)$$

## 2 Proof of Proposition 1

*Proposition 1:* The backward pass has a fixed point of $\zeta^\star = -\mathbb{W}_{-1}(-\exp(-1 - \frac{\delta}{\tau_0})) - 1 - \frac{\delta}{\tau_0}$.

If a fixed point exists, then it must satisfy the following equation:

$$\zeta^\star = \ln\left(1 + \tfrac{\delta}{\tau_0} + \zeta^\star\right) \tag{4}$$

$$\exp(\zeta^\star) = 1 + \tfrac{\delta}{\tau_0} + \zeta^\star \tag{5}$$

$$1 = (1 + \tfrac{\delta}{\tau_0} + \zeta^\star)\exp(-\zeta^\star) \tag{6}$$

$$-1 = (-1 - \tfrac{\delta}{\tau_0} - \zeta^\star)\exp(-\zeta^\star) \tag{7}$$

$$-\exp(-1 - \tfrac{\delta}{\tau_0}) = (-1 - \tfrac{\delta}{\tau_0} - \zeta^\star)\exp(-\zeta^\star)\exp(-1 - \tfrac{\delta}{\tau_0}) \tag{8}$$

$$-\exp(-1 - \tfrac{\delta}{\tau_0}) = (-1 - \tfrac{\delta}{\tau_0} - \zeta^\star)\exp(-1 - \tfrac{\delta}{\tau_0} - \zeta^\star). \tag{9}$$

However, $y = x \exp(x)$ is equivalent to $x = \mathbb{W}(y)$, so

$$(-1 - \tfrac{\delta}{\tau_0} - \zeta^\star) = \mathbb{W}(-\exp(-1 - \tfrac{\delta}{\tau_0})) \tag{10}$$

$$\zeta^\star = -\mathbb{W}(-\exp(-1 - \tfrac{\delta}{\tau_0})) - 1 - \tfrac{\delta}{\tau_0}. \tag{11}$$

There are two branches of the Lambert W function. The lower branch decreases from $\mathbb{W}_{-1}(-\exp(-1)) = -1$ to $\mathbb{W}_{-1}(0) = -\infty$, while the principal branch increases from $\mathbb{W}_0(-\exp(-1)) = -1$ to $\mathbb{W}_0(0) = 0$ and beyond. Because $\zeta^\star$ must be positive, we therefore have

$$\zeta^\star = -\mathbb{W}_{-1}(-\exp(-1 - \tfrac{\delta}{\tau_0})) - 1 - \tfrac{\delta}{\tau_0}. \tag{12}$$

## 3 MCMC Inference

*Definition 1:* If $y_. = \sum_{n=1}^N y_n$, where $y_n \sim \text{Pois}(\theta_n)$ are independent Poisson-distributed random variables, then $(y_1, \ldots, y_N) \sim \text{Mult}(y_., (\frac{\theta_1}{\sum_{n=1}^N \theta_n}, \ldots, \frac{\theta_N}{\sum_{n=1}^N \theta_n}))$ and $y_. \sim \text{Pois}(\sum_{n=1}^N \theta_n)$ [3, 4].

*Sampling the latent subcounts:* Via definition 1,

$$(y_{v1}^{(t)}, \ldots, y_{vK}^{(t)} \mid -) \sim \text{Mult}(y_v^{(t)}, (\frac{\phi_{v1}\theta_1^{(t)}}{\sum_{k_2=1}^K \phi_{vk}\theta_k^{(t)}}, \ldots, \frac{\phi_{vK}\theta_K^{(t)}}{\sum_{k_2=1}^K \phi_{vk}\theta_k^{(t)}})) \tag{13}$$

*Sampling $\Phi$:* Via Dirichlet–multinomial conjugacy,

$$(\phi_k \mid -) \sim \text{Dir}(\eta_0 + \textstyle\sum_{t=1}^T y_{1k}^{(t)}, \ldots, \eta_0 + \sum_{t=1}^T y_{Vk}^{(t)}). \tag{14}$$

*Sampling $\delta^{(1)}, \ldots, \delta^{(T)}$:* Via gamma–Poisson conjugacy,

$$(\delta^{(t)} \mid -) \sim \text{Gamma}(\epsilon_0 + \textstyle\sum_{v=1}^V y_v^{(t)}, \epsilon_0 + \sum_{k=1}^K \theta_k^{(t)}) \tag{15}$$

*Sampling $\delta$ (stationary variant):* Also via gamma–Poisson conjugacy,

$$(\delta \mid -) \sim \text{Gamma}\left(\epsilon_0 + \textstyle\sum_{t=1}^T\sum_{v=1}^V y_v^{(t)}, \epsilon_0 + \sum_{t=1}^T\sum_{k=1}^K \theta_k^{(t)}\right). \tag{16}$$

*Sampling $\beta$:* Via gamma–gamma conjugacy,

$$(\beta \mid -) \sim \text{Gamma}(\epsilon_0 + \gamma_0, \epsilon_0 + \textstyle\sum_{k=1}^K \nu_k). \tag{17}$$

*Sampling $\nu$ and $\xi$:* To obtain closed-form conditional posteriors for $\nu_k$ and $\xi$, we start with

$$(l_{1k}^{(\cdot)}, \ldots, l_{kk}^{(\cdot)}, \ldots, l_{Kk}^{(\cdot)}) \sim \text{DirMult}(l_{\cdot k}^{(\cdot)}, (\nu_1\nu_k, \ldots, \xi\nu_k, \ldots, \nu_K\nu_k)), \tag{18}$$

where $l_{k_1k}^{(\cdot)} = \sum_{t=1}^T l_{k_1k}^{(t)}$ and $l_{\cdot k}^{(\cdot)} = \sum_{t=1}^T \sum_{k_1=1}^K l_{k_1k}^{(t)}$. As noted previously by Zhou [5], when augmented with a beta-distributed auxiliary variable, the Dirichlet–multinomial distribution is proportional to the negative binomial distribution. We therefore draw a beta-distributed auxiliary variable:

$$q_k \sim \text{Beta}(l_{\cdot k}^{(\cdot)}, \nu_k (\xi + \textstyle\sum_{k_1 \neq k} \nu_{k_1})). \tag{19}$$

Conditioned on $q_k$, we then have

$$l_{kk}^{(\cdot)} \sim \mathrm{NB}(\xi \, \nu_k, q_k) \text{ and } l_{k_1 k}^{(\cdot)} \sim \mathrm{NB}(\nu_{k_1} \nu_k, q_k) \tag{20}$$

for $k_1 \neq k$. Next, we introduce the following auxiliary variables:

$$h_{kk} \sim \mathrm{CRT}(l_{kk}^{(\cdot)}, \xi \, \nu_k) \text{ and } h_{k_1 k} \sim \mathrm{CRT}(l_{k_1 k}^{(\cdot)}, \nu_{k_1} \nu_k) \tag{21}$$

for $k_1 \neq k$. We can then re-express the joint distribution over the variables in equations 20 and 21 as

$$l_{kk}^{(\cdot)} \sim \mathrm{SumLog}(h_{kk}, q_k) \text{ and } l_{k_1 k}^{(\cdot)} \sim \mathrm{SumLog}(h_{k_1 k}, q_k) \tag{22}$$

and

$$h_{kk} \sim \mathrm{Pois}(-\xi \, \nu_k \ln(1 - q_k)) \text{ and } h_{k_1 k} \sim \mathrm{Pois}(-\nu_{k_1} \nu_k \ln(1 - q_k)). \tag{23}$$

Then, via gamma–Poisson conjugacy,

$$(\xi \mid - \setminus \Theta, \boldsymbol{\pi}_k) \sim \mathrm{Gam}(\tfrac{\gamma_0}{K} + \textstyle\sum_{k=1}^{K} h_{kk}, \ \beta - \sum_{k=1}^{K} \nu_k \ln(1 - q_k)). \tag{24}$$

Next, because $l_{k\cdot}^{(1)} \sim \mathrm{Pois}(\zeta^{(1)} \tau_0 \nu_k)$ also depends on $\nu_k$, we introduce

$$n_k = h_{kk} + \textstyle\sum_{k_1 \neq k} h_{k_1 k} + \sum_{k_2 \neq k} h_{kk_2} + l_{k\cdot}^{(1)}. \tag{25}$$

Then, via definition 1, we have

$$n_k \sim \mathrm{Pois}(\nu_k \, \rho_k), \tag{26}$$

where

$$\rho_k = -\ln(1 - q_k)(\xi + \textstyle\sum_{k_1 \neq k} \nu_{k_1}) - \sum_{k_2 \neq k} \ln(1 - q_{k_2}) \, \nu_{k_2} + \zeta^{(1)} \tau_0. \tag{27}$$

Finally, via gamma–Poisson conjugacy,

$$(\nu_k \mid - \setminus \Theta, \boldsymbol{\pi}_k) \sim \mathrm{Gam}(\frac{\gamma_0}{\beta} + n_k, \beta + \rho_k). \tag{28}$$

## 4   Additional Results

*Polyphonic music (PM):* To test the models' predictive performance for binary observations, we used a polyphonic music sequence data set [6]. We created binary matrices for the first forty-four songs in this data set. Each matrix has one column per time step and one row for each key on the piano—i.e., $T = 111$ to $3{,}155$ and $V = 88$. A single observation $b_v^{(t)} = 1$ means that key $v$ was played at time $t$.

For each matrix, we created six masks indicating some subset of columns (10%, 20%, and 30%, randomly selected) to treat as held-out data. We only tested the models' smoothing performance because the last time steps are often empty. For each matrix, mask, and model combination, we ran inference and imputed the missing data, as described in section 4 of the main paper. For both the PGDS and GP-DPFA, we used the Bernoulli–Poisson distribution to link the observations to latent Poisson counts [2]. We applied the LDS directly; despite being misspecified, the LDS often performs well for binary data [7]. We used AUC [8] to compute the models' smoothing scores (higher is better) for each inference run, and

Table 1: Smoothing scores (higher is better) for the binary matrices.

| | | AUC | | |
|---|---|---|---|---|
| | Mask | LDS | GP-DPFA | PGDS |
| PM | 10% | **0.94** | 0.92 | 0.93 |
| PM | 20% | **0.93** | 0.91 | **0.93** |
| PM | 30% | **0.93** | 0.91 | 0.92 |

then averaged the scores over the runs, the masks with the same percentage of held-out data, and the matrices. We report our results in table 1. For the LDS, we only report the scores for the best-performing value of $K$. For all three models, performance degraded as we held out more data. The LDS and the PGDS, both of which have an expressive transition structure, outperformed GP-DPFA.

Table 2: Results for the smoothing ("S") and forecasting ("F") tasks using all the features. Lower values are better. We also report the number of time steps $T$ and the burstiness $\hat{B}$ of each data set.

| | $T$ | $\hat{B}$ | Task | Mean Relative Error | | Mean Absolute Error | |
|---|---|---|---|---|---|---|---|
| | | | | PGDS | GP-DPFA | PGDS | GP-DPFA |
| GDELT | 365 | 1.71 | S | **0.428** ±0.06 | 0.617 ±0.06 | **1.491** ±0.22 | 1.599 ±0.21 |
| | | | F | **0.432** ±0.09 | 0.494 ±0.08 | **1.224** ±0.19 | 1.263 ±0.21 |
| ICEWS | 365 | 1.26 | S | **0.334** ±0.02 | 0.372 ±0.01 | **1.003** ±0.13 | 1.021 ±0.14 |
| | | | F | **0.299** ±0.05 | 0.313 ±0.05 | **0.646** ±0.13 | 0.673 ±0.14 |
| SOTU | 225 | 1.49 | S | **0.216** ±0.00 | 0.226 ±0.00 | **0.365** ±0.00 | 0.374 ±0.00 |
| | | | F | 0.172 ±0.00 | **0.169** ±0.00 | 0.295 ±0.00 | **0.289** ±0.00 |
| DBLP | 14 | 1.73 | S | 0.370 ±0.00 | **0.356** ±0.00 | 0.604 ±0.00 | **0.591** ±0.00 |
| | | | F | **0.370** ±0.00 | 0.408 ±0.00 | **0.778** ±0.00 | 0.790 ±0.00 |
| NIPS | 17 | 0.89 | S | 2.133 ±0.00 | **1.199** ±0.00 | 9.375 ±0.00 | **7.893** ±0.00 |
| | | | F | 1.173 ±0.00 | **0.949** ±0.00 | 15.065 ±0.00 | **12.445** ±0.00 |