[Reviews · NeurIPS 2016]

Reviewer 1

Summary

This paper presents a very interesting time series model based on gamma-distributed latent states that evolve under a Dirichlet transition matrix to produce Poisson observations. This expands the state of the art by providing a dynamical system with transitional structure and the benefits of sparsity and efficiency that the Poisson-Gamma construction brings.

Qualitative Assessment

Technical quality: -the paper is strong technically, with interesting alternative parameterizations to make inference closed-form -a full suite of experiments are given looking at various metrics (RMSE, MAE, MRE) and data sets Originality: the approach builds on a previous model [Acharya et al. 2015] to include transition dynamics. Impact: this model/inference method is likely to be used and extended by others and is an additional tool for modeling sparse and bursty time series data. Clarity: overall, the approach is well presented, however there are parts which are quite dense and hard to digest, especially the "Bayesian Non-Parametrics" subsection in Section 2. Ideally, that section would be expanded and made more reasonable, while some of the technical motivations in Section 3 could be relegated to the appendix. Finally, a note on the use of certain vague words (e.g., "powerful" on line 3, "elucidates" on line 142), it would be more comprehensive if they were replaced with more descriptive words.

Confidence in this Review

2-Confident (read it all; understood it all reasonably well)


Reviewer 2

Summary

This paper presents a linear dynamical system (Kalman-style) model for count data. Inference is done via MCMC.

Qualitative Assessment

This is a nice model; I'm surprised that it has not been done before.

Confidence in this Review

2-Confident (read it all; understood it all reasonably well)


Reviewer 3

Summary

The article develops a new factor model for sequentially observed multivariate count data. In detail, the rates of the Poisson random variables are assumed to evolve dynamically according to a transition matrix, similar to linear dynamical systems, which is the main novelty of the approach. The apprach is motivated by requirements of the complex structure of such data, including, for example, the ‘burstiness’, i.e., periods of strong activity of one variable, and complex excitatory relationships between the variables. The rank is let to go to infinity, making the model non-parametric. The conditions of the soundness of the non-parametric prior are theoretically established, and the steady-state properties of the model are investigated theoretically as well. MCMC inference is developed using an alternative generative process based on recently developed techniques for augmentation of the negative-binomial distribution. The experimental section contains a comparison of predictive performance against two alternative models with five real-world data sets, and a nice interpretation of the top components for two of the real-world data sets.

Qualitative Assessment

The new developments are very well motivated. The method is novel, to my knowledge, and interesting. The article is very well written. Additional strengths include the theoretical investigation of the properties of the nonparametric prior, theoretical results about the steady state properties of the model, and the derivation of the non-trivial inference algorithm. The prediction results are not strikingly impressive, but, nevertheless, they show the model has similar or better performance than some alternatives, and is strong in particular when the data exhibit ‘bursty’ characteristics, as expected. The interpretable structure of the learned models was very nicely demonstrated.

Confidence in this Review

2-Confident (read it all; understood it all reasonably well)


Reviewer 4

Summary

This paper proposes a novel model based on Poisson-Gamma construction for modelling multivariate count data. The proposed model addresses some limitation of the existing models (LDS and GP-DPFA), and also incorporate a nonparametric Bayesian extension. This lead to a somewhat advance model and this paper proposes a novel MCMC inference for learning. Experiments show that the proposed model generally outperforms the other existing models.

Qualitative Assessment

The proposed model is novel and practical, as seen from the experimental result. It is rare to see a Bayesian nonparametric model being applied to large data as it is generally not very scalable. It is a feat to see this model applied to data with high dimensions (9000 dimensions with millions of events). I am interested to know how much time is spent for training? It would be good to also present the computational time (say in the supplementary material). Though I did not go through the derivation line by line, the inference methods appear technically sound and will be valuable to other researchers. The forward and backward pass in the inference seems to be related to hidden Markov model (HMM), perhaps some links to the HMM can be elaborated? Experiments are great. However, I have a couple of questions: 1. Gibbs sampler was performed for 10000 iterations but only 60 samples are used. Is the number of samples enough, since only 6% of the samples are used? 2. Why is the result in table 1 only uses V=500? But full result is presented in the supplementary material instead? Figure 1 shows interesting result and thus I believe this model will be useful in other disciplines. Will the code for this model released publicly? Some minor comments: 1. Shrinking the font size of the reference list will make it more readable. 2. Reference [3] has the main author replicated 3 times. 3. Some proper nouns in the references are not capitalized, e.g. Poisson, Bayesian... 4. Would be nice to insert the acronym "CRT" after "Chinese Restaurant Table" in page 4. i.e. "Chinese Restaurant Table (CRT)". As this facilitates searching of the acronym. 5. Citation [9] is used as noun, will be much readable to include the authors before [9]. i.e. "taken by Acharya et al [9]" in page 2.

Confidence in this Review

2-Confident (read it all; understood it all reasonably well)


Reviewer 5

Summary

The authors propose a Bayesian estimation method for Poisson-Gamma dynamical systems, using Bayesian nonparametric priors and MCMC algorithms. The proposed approach can outperform other methods in modeling multivariate dynamic count data.

Qualitative Assessment

MAJOR ISSUES 1. Sections/paragraph/subparagraph numbering is very confusing. For instance, the titles of the subsections/paragraphs in lines 78 to 130 should be numbered. Similar comment on Section 3 MCMC: it seems that "Negative binomial distribution" and "Alternate generative process" are the main subsections (e.g. with numbers 3.1 and 3.2), but what about "Steady state" from line 183 to 197? Does it refer to both the subsections or only the "Alternate generative process"? Similar comment on the algorithm between rows 182 and 183. This algorithm should be formally presented as "Algorithm 1" in algorithm style. Similarly, lines 198 to 216 describe another algorithm that could be reorganized as "Algorithm 2", or at least a subsection or paragraph title (e.g. 3.3) should be written before line 198. Furthermore, rows from 229 to 267 seems to constitute a subsection, which might be called "4.1 Count data", and similarly rows 268 to 283 could represent a subsection called "4.2 Binary data". Finally rows 284 to 318 should also be splitted into subsections. 2. How the use of the "alternative generative process" compares with "the negative binomial distribution" in terms of computational times? MINOR ISSUES 1. This is a highly technical paper that, in general, might be improved by moving most of the details into an appendix or the supplementary material. 2. On row 75 and in the rest of the paper, it is not specified what 1_V means. 3. In Table 1, are the results mean +- 2sd or just 1sd? If it is only 1sd, please try to replace it with 2sd.

Confidence in this Review

2-Confident (read it all; understood it all reasonably well)


Reviewer 6

Summary

This paper considers modelling sequential count data, which arises in many real world applications but results in challenges in its analysis due to its high dimensionality, overdispersion and potentially non-linear dynamics governing its structure over time. The authors develop a novel Gamme-Poisson dynamical system (PGDS), assuming a latent sequence of Gamma variables and Poisson observations. They impose shrinkage priors that allow for automatic tuning of the latent state dimensionality and extend the model to a non-parametric approach, defining two novel stochastic processes. For inference in this model, the authors develop an MCMC algorithm with efficient sampling updates, making use of two equivalent generative processes of the model and auxiliary variables. The authors perform a series of experiments on different datasets, comparing the PGDS to the classic Gaussian linear dynamical system and the Gamma process dynamic Poisson Factor Analysis, showing that the PGDS outperforms the other two models in a number of applications, in particular in cases with overdispersed data. The authors further illustrate the interpretability of the learned model parameters in regards to real world applications.

Qualitative Assessment

Technical quality: This is a rigorous and well-written paper with a lot of technical detail. The details included in the main paper are sufficient to develop a general understanding of the methods used, with the supplementary material offering proofs and more details on the sampling scheme. The experiments show the model's applicability to a variety of real world datasets, which ties in nicely with the way the method was motivated. Minor details: the introduction mentions an Extended Kalman filter as a non-Gaussian extension to the LDS. I think this generally refers to non-linear extensions of the LDS, rather than non-Gaussian ones. There is also a larger body of work in the neuroscience literature that has been concerned with extensions of the LDS to Poisson observations (PLDS), combining the expressive power of Gaussian latent dynamics with Poisson observations. This could be included in the literature review and might be a nice contrast to the approach presented in this paper. Novelty/originality: The paper makes notable novel contributions by introducing the PGLD together with novel stochastic processes. Potential impact or usefulness: Novel approaches to modelling sequential count data should be applicable to a variety of real world processes, such as social networks or Neuroscience. Clarity and presentation: The paper is well-written and coherent. At points there are small errors (e.g. line 76) or typos which would require corrections, but these are very minor.

Confidence in this Review

2-Confident (read it all; understood it all reasonably well)